# Changing Patterns of the Flow Ratio with the Distance of Exhaust and Supply Hood in a Parallel Square Push-Pull Ventilation

**DOI:** 10.3390/ijerph19052957

**Published:** 2022-03-03

**Authors:** Jianwu Chen

**Affiliations:** 1School of Civil and Resource Engineering, University of Science and Technology Beijing, Beijing 100083, China; cjw3000@126.com; Tel.: +86-10-6494-1249; 2Institute of Occupational Health, China Academy of Safety Science and Technology, Beijing 100012, China

**Keywords:** push-pull ventilation, flow ratio, control distance, parallel flow

## Abstract

The method of flow ratio *k* is often used for designing parallel push-pull ventilation. The *k* value is mostly selected empirically and is difficult to determine accurately, resulting in an imprecise design of the push-pull ventilation system. Therefore, parallel push-pull ventilation was taken as the research object in this paper. The push-pull ventilation studied consists of a square uniform supply hood and a square uniform exhaust hood, and the side length of pull hood and pull hood was same. A workbench was set between the push hood and pull hood, and the source of toluene pollutions was set in the center of the worktable surface. The optimal *k* values for different distances between push hood and pull hood were studied by numerical simulation using Ansys Fluent, which were obtained base on the distribution of wind speed and toluene concentration. The results showed that parallel push-pull ventilation is not suitable for applications when *L*/*a* ≥ 6. The changing patterns of *k* value with *L*/*a* is proposed in the range of 1.5 ≤ *L*/*a* ≤ 5 for the parallel square push-pull ventilation, which can be used to estimate *k* value relatively accurately under the condition that *L*/*a* is known, so as to guide the determination of the exhaust air volume of the parallel push-pull ventilation system.

## 1. Introduction

Ventilation is an important methods to control contaminants in workplaces [1,2]. Compared with local exhaust ventilation, push-pull ventilation has better contaminants control due to increased air supply [3,4]. Parallel push-pull ventilation uses the low turbulence intensity, uniform and wide air flow with directionality to push the contaminants into the exhaust outlet [5,6], which has been widely used in workplace contaminants control such as painting [7,8], printing [9] and cleaning using organic solvent [10,11], welding [12] and pathology laboratories [13].

Wu X. et al. studied the influence of a 90-degree elbow on the velocity uniformity of exhaust hoods in parallel push-pull ventilation [14]. Chen J. et al. studied the internal structure of the static pressure chamber in a spray room in order to get the uniform air supply [15], and also studied the center-line velocity change regime in a parallel-flow exhaust hood [16] and supple hood [17]. Wang Y. discussed the effects of the pull-flow velocity on the capture efficiency in a high-velocity jet push-pull ventilation system [18] and found that the significant reduction in exhaust air flow ratio will not affect the dispersion of contaminants in parallel push-pull ventilation systems [19]. However, if the air flow ratio is too small, the contaminants may not be completely trapped by the exhaust hood. Therefore, it is very important to choose a proper flow ratio, which is not only effectively capturing contaminants, but also saves energy.

In order to achieve the best performance of parallel push-pull ventilation, a proper flow ratio should be chosen. The design manual [20] provides an empirical value of flow ratio for the design of parallel push-pull ventilation systems, but it is difficult to design the proper flow ratio in practice of push-pull ventilation design. The optimal air flow ratios of parallel push-pull ventilation system were studied in a washing room [10,11] and a vertical push-pull system [21], but the application of the results is limited to the specific distance conditions of the push-pull ventilation system. When the conditions change, the results cannot be used. The main purpose of this paper is to put forward the change patterns of the flow ratio with the distance of exhaust and supply hood. The research results provide the basis for the accurate determination of the flow ratio in the design of the parallel push-pull ventilation system.

## 2. Subjects and Methods

### 2.1. Geometric Models

A schematic diagram of the geometric model drawn by CAD for this study is shown in Figure 1.

If there is only uniform supply air [16] for square hood, the center-line velocity and *L_off push hood_*/*a* have good change patterns, and if there is only uniform exhaust air [17] for square hood, the center-line velocity and *L_off pull hood_*/*a* also have good change patterns; however, the variation of wind speed with distance is different in the above two cases. Compared with other geometric parameters of exhaust hood, such as short side, long side and square root of area of exhaust hood, the center line with *L_off pull hood_*/*d* have the best change patterns for a desktop slot exhaust hood [1], where *d* is the equivalent diameter of hood face. It is not clear whether the center-line velocity has a better correlation with *L*/*a* or *L*/*d* after the uniform supply air and uniform exhaust air are combined to form a parallel push-pull ventilation. In order to avoid the influence of the geometric structure on the study results, square hoods are used in this study, and the side length of hoods is consistent with the references [18,19]. Therefore, a square push hood and a square pull hood with side length of 0.7 m are built, respectively, and a 0.7 m wide workbench is placed between push hood and pull hood. The distance between the push hood and push hood (*L*) is the same as the length of the workbench. A 0.3 m long × 0.3 m wide × 0.05 m high toluene emitting source was established at the center of the workbench.

The numerical simulation of this paper is carried out by Ansys Fluent. The common building height of 3.5 m is used as the height of the calculation domain. If the distance between the supply hood and exhaust hood is too far in the push-pull ventilation, the exhaust velocity will be very high, which will result in a significant increase in energy consumption, so 8 times the side length of the exhaust hood is selected as the length and width of the calculation domain for this study. Therefore, a calculation domain of 5.6 m long × 5.6 m wide × 3.5 m high is established. The mesh method of using Tet/Hybrid will increase the computation time, but the mesh quality and adaptability is better, so meshing was used in ICEM, and the local grid was encrypted. The accurate grid division was ensured through the grid independence test.

### 2.2. Research Conditions

It is assumed that the air in this study is an incompressible fluid and only momentum transfer is considered, neglecting heat transfer and viscosity between fluid molecules, and these assumptions coincide with the *k*-*ε* model; therefore, the 3D steady-state incompressible Navier–Stokes equations and the standard *k*-*ε* equation model are used in this study, and the flow field of the push-pull ventilation system is solved using the 3D single-precision solver.

In order to avoid the effect of the internal structure of the hood on the uniformity of airflow, the push hood face and the pull hood face were set as Inlet and Outlet, respectively, in this study, and their boundary conditions were both set as Velocity Inlet. The push and pull hoods were both set as square exhaust hoods with side length of 0.7 m, and their area is also same. Therefore, the air volume ratio of exhaust to supply is equal to the wind speed ratio. The push hood face velocity (*v*_1_) is set as 0.5 m/s, and the pull hood face velocity (*v*_2_) is *k* times that of the push hood face, where *k* is the flow ratio of exhaust air volume to supply air volume. The turbulence intensity (*I*) of the velocity inlet is 4.55% calculated by Equation (1).
(1)$$I=0.16Re^{\left(-1/8\right)}$$
where *Re*—Reynolds number, calculated by Equation (2).
(2)Re=ρ·v·d/η
where:*ρ*—gas density, kg/m^3^. The gas density of air is 1.205 kg/m^3^ at 20 °C used in this study.*v*—hood face velocity, m/s. See Table 1.*d*—hydraulic diameter, m. It is 0.7 m for the push hood and pull hood in this study.*η*—dynamic viscosity, Pa·s. The dynamic viscosity of air is 1.81 × 10^−5^ Pa·s at 20 °C used in this study.


The values of pull hood face velocity, turbulence intensity and Reynolds number of outlet at different flow ratios are shown in Table 1. The turbulence intensities (*I*) of the velocity outlet are calculated by Equation (1), and the Reynolds numbers are calculated by Equation (2).

The toluene emitting area is 0.3 m × 0.3 m, the toluene emitting volume is 500 mg/s, and the mass fraction is 0.9. Except for the push hood face, the pull hood face and the toluene emitting source, the rest of surface in the model are set to Wall, and the boundary conditions and solver settings are shown in Table 2.

## 3. Results

Push-pull ventilation can significantly increase the contaminants control distance of the local exhaust system [3,4]. Thus, this study uses a distance (*L*) of 1.5 times the side length of the hood as the starting research condition, and studies it until the maximum distance at which the contaminants cannot be effectively controlled. The optimal *k* values were investigated for six distances (*L*/*a* = 1.5, 2, 3, 4, 5, 6) in this paper.

### 3.1. Results of the Optimal k Value at Different Distances (L/a)

The distance (*L*) from the push hood to the pull hood is set as 1.05 m, at which time *L*/*a* is 1.5. The contaminants, supply air and disturbing airflow should be push into the pull hood, so the exhaust air volume should be no less than the supply air volume. Therefore, take *k* = 1 as the initial condition when *L*/*a* = 1.5. The push-pull ventilation system is numerically calculated by adjusting the exhaust air volume for different *k* values (*k* = 1, 1.2, 1.5, 1.7, 2) when *L*/*a* = 1.5. The results of the wind speed and toluene concentration distribution studied are shown in Figure 2.

As can be seen from Figure 2, when *k* = 1, a large amount of toluene escapes to the outside of the pull hood, mainly due to the small value of *k* (i.e., small exhaust air volume), and the exhaust air volume cannot draw the contaminants into the pull hood. As the *k* value increases, the concentration of toluene escaping to the outside of the pull hood decreases rapidly, and when *k* increases to 1.5, the concentration of toluene basically does not spread to the outside of the exhaust hood, mainly due to the increase the exhaust air volume to enough to completely capture the contaminants into the pull hood. It can be seen from Figure 2 that when the *k* value is greater than 1.5, with the increase in *k* value the wind speed control distance at the pull hood becomes larger and larger, but there is no significant change in the control performance of toluene. It means that when *k* = 1.5 can meet the requirement of toluene concentration control, and further increase will only increase the energy consumption, and the improvement of toluene control performance is not significant.

The main purpose of push-pull ventilation is to protect workers’ occupational health, so it is necessary to analyze the toluene concentration at the location of workers’ breathing zone at different *k* values when *L*/*a* = 1.5. In general, workers stand at a distance of about 0.2 m outside the worktable to work, while the breathing zone is normally located at a height of 1.5 m from the ground. Therefore, the toluene concentration 0.2 m outside the worktable and 1.5 m high above the ground parallel to the horizontal line of the ventilation system is taken as the concentration of toluene in workers’ breathing zone. According to the results of numerical simulation, the toluene concentration at the middle position of the worker’s breathing zone for different *k* values at *L*/*a* = 1.5 is shown in Figure 3.

As shown in Figure 3, the toluene concentration is very large when *k* = 1, and the toluene concentration decreases rapidly as the *k* value increases before the *k* value reaches 1.5; when *k* value increases to 1.5, the toluene concentration is close to 0 mg/m^3^, and then increasing the *k* value has basically no effect on the toluene concentration, which is consistent with the results of Figure 2. It can be seen that when *L*/*a* = 1.5, toluene concentration can be effectively controlled when *k* = 1.5, and increasing the value of *k* only increases the energy consumption of the system and has very little effect on toluene concentration, so the optimal *k* value is 1.5 at *L*/*a* = 1.5.

The effect of different *k* values on toluene concentration was studied separately for the other four conditions of *L*/*a* (2, 3, 4, 5), using the same study method described above, and the optimal *k* values were obtained, where the initial study conditions of *k* values were determined with reference to the previous set of optimal *k* values. The optimal *k* values studied for the five conditions of *L*/*a* (1.5, 2, 3, 4 and 5) are shown in Table 3.

### 3.2. Results of the Longest Control Distance (L/a) for Parallel Square Push-Pull Ventilation

In order to obtain the longest control distance for parallel square push-pull ventilation, the distance (*L*) was set to 4.2 m, i.e., the case at *L*/*a* = 6 was investigated. Combined with the results of the previous numerical simulation analysis, the *k* value should gradually increase as *L*/*a* increases. Thus, when *L*/*a* = 6, a value slightly lower than the optimal *k* value of 4.3 at *L*/*a* = 5 is taken as the initial study condition, and when *k* value increases to 10, the exhaust air volume is 10 times the supply air volume, which is basically unlikely in practical applications, so the upper limit of *k* value is set to 10 in this study. Using the same numerical simulation method, the numerical simulations were carried out for different *k* values (*k* = 4.3, 5, 8, 10) at *L*/*a* = 6. The results of the wind speed and toluene concentration distribution studied for different *k* values at *L*/*a* = 6 are shown in Figure 4.

Figure 4 showed that the exhaust air volume is too small when *k* = 4.3, resulting in a large amount of toluene escaping, and with the increase in *k* value, the toluene concentration decreases rapidly, but the more the pull hood draws in the surrounding air, even when *k* increases to 10, there is still a large amount of toluene escaping. According to results studied, the toluene concentrations at workers’ breathing zone for different *k* values at *L*/*a* = 6 were plotted as Figure 5.

With the increase in *k* value, the maximum value of toluene concentration gradually becomes smaller, and the distribution range of toluene in the horizontal direction of the table gradually becomes smaller. When *k* increases to 10, toluene still escapes to the outside of the pull hood, and the highest toluene concentration in front of the pull hood is still as high as 88.9 mg/m^3^, far greater than the standard requirement of 50 mg/m^3^ occupational hazard limit value. However, the *k* value is far beyond the range of “2–5 when *L*/*a* is 4–5 or more”, as specified in the design manual [20], and at this time, the exhaust air volume is too large. The actual application is basically unlikely to apply because of the energy consumption and other issues. Therefore, the parallel push-pull ventilation is not suitable for applications when *L*/*a* ≥ 6.

### 3.3. Changing Patterns of k Value with L/a When 1.5 ≤ L/a ≤ 5

From the results studied, it is clear that parallel push-pull ventilation is suitable for use at *L*/*a* ≤ 5, when the contaminants are located the center of the push and pull hoods. The optimal *k* values in Table 2 at different *L*/*a* were plotted by Excel and analyzed by curve fitting. The results are shown in Figure 6.

In the range of 1.5 ≤ *L*/*a* ≤ 5.0, a polynomial was used to fit the relationship between *k* and *L*/*a*. When a second-order polynomial was used to fit, *R*^2^ = 0.9719. When a third-order polynomial was used to fit, *R*^2^ = 0.9989. It indicated that the results of fitting with a third-order polynomial have good reliability. The relationship between *k* and *L*/*a* is shown in Equation (3).
*k* = 0.1675(*L*/*a*)^3^ − 1.318(*L*/*a*)^2^ + 3.6085(*L*/*a*) − 1.5352(3)

Yang et al. [10] and Liang et al. [11] found that the optimal *k* value is 2.5 in a push-pull ventilation system, which consists of a uniform push hood and a uniform pull hood with a side length of 0.95 m, and the distance (*L*) between push hood and pull hood is 3.5 m in the push-pull ventilation studied, so *L*/*a* = 3.7. The optimal *k* value is about 2.25, calculated by Equation (3) proposed in this paper, which is about −11% deviation of the calculated *k* value from the results studied by Yang et al. [10] and Liang et al. [11]. It indicated that the variation patterns of *k* value with *L*/*a* when 1.5 ≤ *L*/*a* ≤ 5 proposed in this study is plausible.

## 4. Discussion

The optimal *k* value calculated by Equation (3) proposed in this paper was smaller than that of the results found by Yang et al. [10] and Liang et al. [11], which may be due to the smaller ratio of contaminants width to hood width in this study. The ratio of contaminants width to hood width is 0.43 in this study, as the contaminants width is 0.3 m and the hood width is 0.7 m. While the ratio of contaminants width to hood width is 0.83 in Yang’s [10] study and Liang’s [11] study, as the contaminants width is 0.8 m and the hood width is 0.95 m. Therefore, the larger the ratio of contaminants width to hood width, the closer the contaminants are to the edge of the worktable, the more difficult it is to control the contaminants, and the larger the exhaust air volume needed to control the contaminants, i.e., the larger the *k* value should be. Therefore, when the ratio of contaminants to hood width is different to that of this study, it is necessary to modify Equation (3) to calculate the *k* value for the design of a parallel push-pull ventilation system.

As shown in Figure 4, when *L_off push hood_*/*a* is about 5 and *L_off pull hood_*/*a* is about 1 in the parallel push-pull system, the toluene concentration is highest at this position. According to Reference [16], if there is only uniform supply air, the supply velocity at this position is about 90% of the push hood face velocity. According to Reference [17], if there is only uniform exhaust air, the exhaust velocity at this position is about 10% of the pull hood face velocity. As can be seen from Figure 5, the velocity at this position is about 0.5 m/s, which is basically the same as the push hood face velocity, but the toluene concentration is still high even with *k* = 10. This is probably due to the contaminants being too far from the pull hood to control, when the distance between the contaminants and the pull hood is 3 times the side length of the hood face, at which position more interfering air flows into the pull hood, so the contaminants may can be controlled if they are closer to the pull hood, not at the center of the worktable. Therefore, further study on the location of pollutants can be carried out later.

The farthest control distance and the variation law of *k* value with distance in the range of 1.5 ≤ *L*/*a* ≤ 5 for parallel square push-pull ventilation are proposed in this study; however, the following aspects should be noted in the application of the results of this study.

(1)The results of this study are based on the square hood. If the hood is circular or rectangular, it may be estimated by equivalent diameter according to the results in Reference [1] and modified appropriately.(2)There is a worktable between the push and pull hood without flange. If a change is found in the model, the results of the study should be applied with appropriate modifications.(3)This study does not consider the influence of disturbing the air flow, but it may exist in practice. The size and position of contaminants in practice may have some deviation from the model in this study. Therefore, the results of this study should be applied with appropriate modifications in those above cases.

## 5. Conclusions

(1)When *L*/*a* is 6 or more, a parallel push-pull ventilation is not suitable for use.(2)The relationship between *k* and *L*/*a* for a parallel push-pull ventilation can be expressed in the equation of *k* = 0.1675(*L*/*a*)^3^ − 1.318(*L*/*a*)^2^ + 3.6085(*L*/*a*) − 1.5352, when 1.5 ≤ *L*/*a* ≤ 5.0.(3)If *L*/*a* is known, a relatively accurate *k* value can be determined by using Equation (3), presented in this study, which can be used in the design of a parallel push-pull ventilation system. When the actual situation and the conditions of this study are different, the appropriate amendments should be made.

## Figures and Tables

**Figure 1 ijerph-19-02957-f001:**
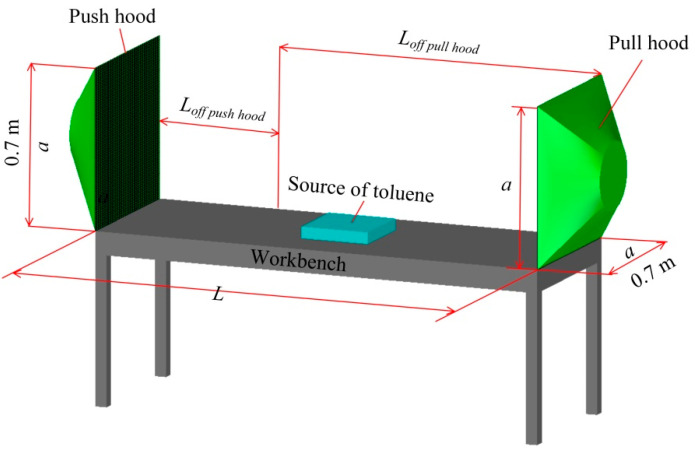
Geometric model of a push-pull ventilation system. *a*: side length of square hood, which is 0.7 m in this study, *L*: the distance between push hood and pull hood, *L_off push hood_*: the distance from push hood, *L_off pull hood_*: the distance from pull hood.

**Figure 2 ijerph-19-02957-f002:**
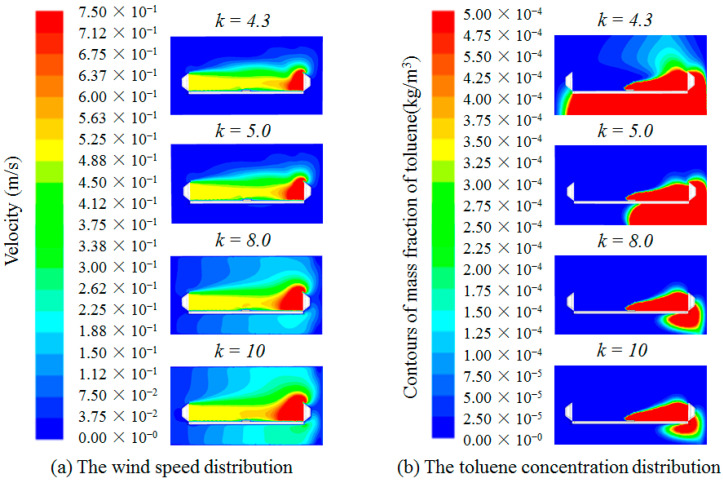
Distribution of wind speed and toluene concentration for different *k* values at *L*/*a* = 1.5.

**Figure 3 ijerph-19-02957-f003:**
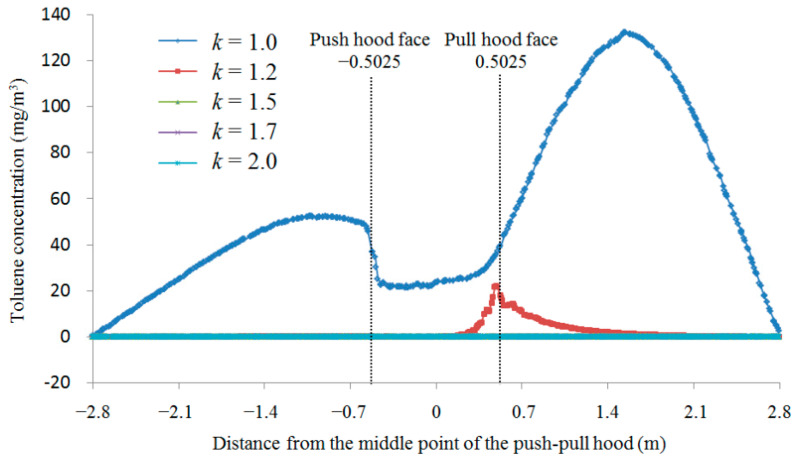
Toluene concentration at respiratory zone locations for different *k* values at *L*/*a* = 1.5. The data lines of *k* = 1.5, 1.7, and 2.0 overlap.

**Figure 4 ijerph-19-02957-f004:**
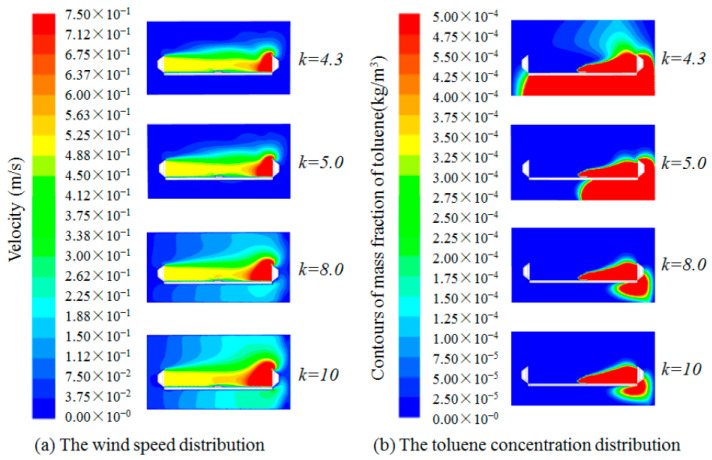
Distribution of wind speed and toluene concentration for different *k* values at *L*/*a* = 6.

**Figure 5 ijerph-19-02957-f005:**
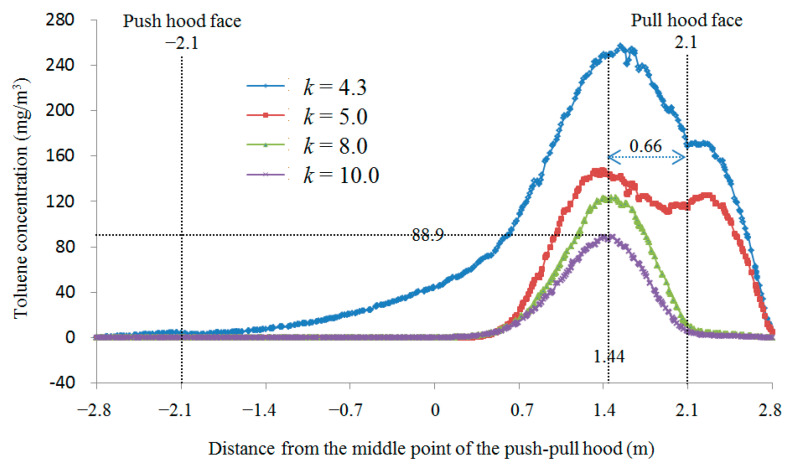
Toluene concentration at respiratory zone locations for different *k* values at *L*/*a* = 6.

**Figure 6 ijerph-19-02957-f006:**
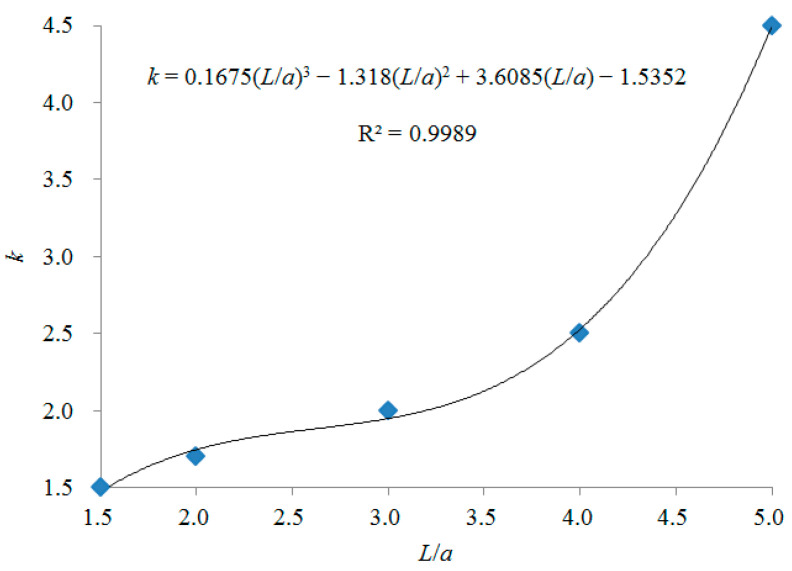
The variation patterns of *k* value with *L*/*a* when 1.5 ≤ *L*/*a* ≤ 5.

**Table 1 ijerph-19-02957-t001:** The values of pull hood face velocity, turbulence intensity and Reynolds number of outlet at different flow ratios.

Push Hood Face Velocity (*v*_1_, m/s)	Flow Ratio (*k*)	Pull Hood Face Velocity (*v*_2_*,* m/s)	Turbulence Intensityof Outlet (*I*, %)	Reynolds Number of Outlet (*Re*)
0.50	1.0	0.50	4.55%	2.33 × 10^4^
0.50	1.2	0.60	4.45%	2.80 × 10^4^
0.50	1.5	0.75	4.33%	3.50 × 10^4^
0.50	1.7	0.85	4.26%	3.96 × 10^4^
0.50	2.0	1.00	4.17%	4.66 × 10^4^
0.50	2.2	1.10	4.12%	5.13 × 10^4^
0.50	2.3	1.15	4.10%	5.36 × 10^4^
0.50	2.5	1.25	4.06%	5.83 × 10^4^
0.50	2.7	1.35	4.02%	6.29 × 10^4^
0.50	3.0	1.50	3.97%	6.99 × 10^4^
0.50	3.1	1.55	3.95%	7.22 × 10^4^
0.50	3.6	1.80	3.88%	8.39 × 10^4^
0.50	4.0	2.00	3.83%	9.32 × 10^4^
0.50	4.3	2.15	3.79%	1.00 × 10^5^
0.50	4.5	2.25	3.77%	1.05 × 10^5^
0.50	5.0	2.50	3.72%	1.17 × 10^5^
0.50	8.0	4.00	3.51%	1.86 × 10^5^
0.50	10.0	5.00	3.41%	2.33 × 10^5^

**Table 2 ijerph-19-02957-t002:** Boundary conditions and solver parameters.

Boundary Conditions	Define
Inlet	Push hood face
Inlet boundary type	Velocity-inlet
Velocity inlet (m/s)	0.5
Material	air
Air viscosity (kg/(m·s))	1.81 × 10^−5^
Hydraulic diameter of inlet (m)	0.7
Turbulence intensity of inlet (%)	4.55
Outlet	Pull hood face
Outlet boundary type	Velocity-inlet
Velocity outlet (m/s)	See Table 1
Turbulence intensity of outlet (%)	See Table 1
Species mass fractions	C7H8 0.9
Mass flow rate (mg/s)	500
Species transport	On
Solver type	Pressure-based
Solver velocity formulation	Absolute
Solver time	Steady
Viscous model	standard *k*-*ε*
Energy	On
Pressure-velocity coupling scheme	SIMPLEC
Discrete format	Second order upwind
Convergence criterion	10^−6^
Interaction to plot and store	1000

**Table 3 ijerph-19-02957-t003:** Results of the optimal *k* values studied at five different *L*/*a*.

No.	*L*/*a*	*k* Value Research Range	Optimal *k* Value
1	1.5	1.0, 1.2, 1.5, 1.7, 2.0	1.5
2	2.0	1.0, 1.5, 1.7, 2.0, 3.0	1.7
3	3.0	1.5, 1.7, 1.9, 2.0, 2.3, 2.7	2.0
4	4.0	1.5, 2.2, 2.3, 2.5, 2.7, 3.0	2.5
5	5.0	2.7, 3.1, 3.6, 4.0, 4.5, 5.0	4.5

## Data Availability

The data presented in this study are available on request from the corresponding author. The data are not publicly available due to privacy or ethical.

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
