# Peer review of "Changing Patterns of the Flow Ratio with the Distance of Exhaust and Supply Hood in a Parallel Square Push-Pull Ventilation"

_ijerph, 2022, doi:10.3390/ijerph19052957_

Round 1

Reviewer 1 Report

1 The range of color bar in Fig.2 and Fig.4 is not appropriate, leading to an unclear distribution of velocity field and  mass fraction.

2  The line of K =1.5,1.7 and 2.0 is coincided with  x-axis. It is suggested to adjust the range of y-axis.

3  It is suggested to replace the "x" and "y" of the third-order polynomial with "K" and "L/a".

Author Response

Thank you very much for your review. We have revised the paper according to the expert's opinions. Please see the text in detail.

Reviewer 2 Report

The paper presents the results of numerical simulation of the uniform airflow push pull ventilation. The same size square supply and exhaust hoods are applied, and optimal value of flow ratio k is searched for different values of ratio L/a. The authors inform that the k value is mostly selected empirically and is difficult to determine accurately, resulting in an imprecise design of the push pull ventilation system. However, nothing in the paper could help in k precise determination, because the authors tell nothing about it. The whole paper is really just a presentation of some numerical simulation, and the article has serious flaws:

The title does not correctly inform about the topic of the research -  there is no changing pattern of the exhaust and supply air volume ratio, there are rather different patterns of air velocity and toluene concentration changing with different values of k and supply-exhaust hood spacing L/a.

The abstract corresponds to the contents of the article but the last statement of it is not true.

The introduction does not provide a sufficient overview of the problem, the cited articles are not specified, there is no use of citing research works dated 1985 or 1995 without explaining what important information can be found in them.

The numbering of references must be wrong (line 30).

The definition of L is missing, the information about numerical simulation is insufficient, the results are presented in a confused form. The values of turbulence intensity for different velocities in Table 1 are not explained.

The words inlet-outlet are interchanged (line 90, Table 2)

No useful conclusions are presented.

References:

3,9,18,22 the names of authors are missing; the formatting of the references is not uniform

Author Response

(The authors gave the same response as above.)
